# A Student-Centric Evaluation of a Program Addressing Prevention of Gender-Based Violence in Three African Countries

**DOI:** 10.3390/ijerph20156498

**Published:** 2023-08-01

**Authors:** Helen Cahill, Babak Dadvand, Anne Suryani, Anne Farrelly

**Affiliations:** 1Melbourne Graduate School of Education, University of Melbourne, Parkville, VIC 3010, Australia; annef@unimelb.edu.au; 2School of Education, Latrobe University, Bundoora, VIC 3086, Australia; b.dadvand@latrobe.edu.au; 3School of Educational Psychology & Counselling, Faculty of Education, Monash University Peninsula Campus, Frankston, VIC 3199, Australia; anne.suryani@monash.edu

**Keywords:** gender-based violence, prevention education, student voice

## Abstract

Studies investigating the effectiveness of school-related gender-based violence prevention programs seldom report on the extent to which students themselves value and recommend such programs. Yet, along with evidence about effectiveness in relation to shifts in knowledge, attitudes, or intentions, student-valuing is a significant indicator that the programs can make a positive contribution to students’ lives. This mixed-method study analyses survey and focus group data collected from ninety-two schools in three African countries (Tanzania, Zambia, and Eswatini). Students found the program contributed to improved peer relationships and identified the five most useful components as learning about gender equality and human rights, learning how to obtain help for those affected by violence, understanding and communicating about their emotions, strategies to avoid joining in with bullying and harassment, and understanding the effects of gender-based violence.

## 1. Introduction

Studies investigating programs addressing the prevention of school-related gender-based violence (GBV) seldom report on the extent to which students themselves value such programs. Yet, along with evidence about effectiveness in relation to knowledge, attitudes, or intentions, student-valuing is a significant indicator that the programs can make a positive contribution to their lives. Research into program relevance and effectiveness is particularly important for education systems as they seek guidance about program suitability and sustainability. This mixed-method study takes a student-centric approach to the evaluation of the program conducted in ninety-two schools in three African countries (Tanzania, Zambia, and Eswatini), each of which piloted a modified version of the Connect with Respect: Preventing gender-based violence in school (CWR) program [1], which focuses on the prevention of school-related GBV.

The World Health Organization (WHO) urges a comprehensive intersectoral approach to the preventable global problem of GBV [2]. Education systems are urged to ensure that schools provide safe and enabling environments as well as education programs that challenge harmful gender stereotypes, strengthen interpersonal skills, and promote relationships based on equality and consent [3]. School-related GBV sits within the broader social framework of communal gender-based violence.

This imperative to provide prevention education is driven by the recognition of the scale of the social problem. Globally, 27% of ever-married women have experienced physical and/or sexual violence perpetrated by their spouse or intimate partner [2]. Estimates of the prevalence of lifetime intimate partner violence range from 22% of women in high-income countries [4] to 44% in the African region [5]. This form of violence has multiple negative effects on women and their children, including injuries, mental health problems, disruptions to work and schooling, and the intergenerational normalisation of violence [6].

Prevalence data show that rates of GBV tend to be highest in countries where there are also high rates of poverty, acceptance of gender inequity, and where other violence-endorsing attitudes persist [7,8]. Some countries in Africa, including two of the three that are the focus of this study (Zambia and Tanzania), have higher prevalence rates than others. In Tanzania, the life-time prevalence for ever-married or partnered women aged 15–49 is 38% (with 24% experiencing violence in the last year), and for Zambian women, rates are even higher at 41% (with 28% experiencing violence in the last year) [4].

Intimate partner violence and violence against children tend to co-occur within families and to have intergenerational effects [9]. Children who witness their fathers use violence against their mothers not only experience the double negative of bystander trauma and negative role-modelling but are also more likely to experience corporal punishment from parents. This, in turn, contributes to the normalisation of violence, and intergenerational cycles of perpetration and victimisation [7]. Muleneh et al. (2021), in a metanalysis of 50 studies investigating GBV in Sub Saharan Africa, found women with GBV-tolerant views were twice as likely to experience GBV as compared with those who are intolerant of GBV [8]. Where these women with GBV-tolerant views are in communities with similar attitudes, they are 41% more likely to experience spousal violence [8]. High rates of violence-justifying attitudes (ranging from 27% to 49%) are evident in the three countries participating in this study and, as research often finds, in two of the countries (Zambia and Tanzania), women have higher rates of violence-accepting attitudes than men [10].

Adolescent girls are vulnerable to many forms of gender-based violence including sexual violence [11], and gender norms can lead to victims justifying, normalising, or accepting acts of violence [12]. Rates of sexual abuse in adolescent girls are high in Tanzania, Eswatini, and Zambia. Data collected from 18–24 year olds via the Violence Against Children studies conducted in Tanzania and Zambia [13] show that nearly three in ten females (28%) and approximately one in seven males (13%) in Tanzania experienced sexual violence prior to the age of 18. In addition, almost three-quarters of females (74%) and males (72%) experienced physical violence perpetrated by an adult figure prior to 18. Those who experienced sexual violence had higher rates of also experiencing physical violence, with more than eight in ten females and males who experienced sexual violence prior to age 18 also experiencing physical violence from an adult figure. Similar data collected in Zambia show that one in three females and two in five males experienced physical violence prior to age 18, and females were twice as likely (20%) as males (10%) to have experienced sexual abuse prior to age 18 [14]. In Swaziland, these data were collected for girls only. It showed that approximately one in three (33%) females experienced some form of sexual violence prior to age 18 and the same proportion experienced some form of physical violence at the hand of an adult [15]. These studies also demonstrate that it is very rare for young people to seek help in response to sexual abuse. Key reasons for not reporting include fear of abandonment, not being aware that the action constituted abuse, and lack of knowledge about who to tell.

Qualitative research investigating school-related forms of GBV in the African context finds that it can take multiple forms, including verbal, physical, psychological, and sexual, and can be more likely to take place in the unsupervised contexts of the school yard, or during travel to and from schools. Moma’s study of girls’ experiences within South African schools found the playground to be a site in which young people re-enacted the power dynamics witnessed in their sociocultural worlds [16]. The 12–13-year-old girls in her study reported that it was common to encounter sexualised and coercive talk about girls’ bodies in the schoolyard [16]. This less-supervised space also provided opportunity for violence perpetration to occur and to escape follow up. Leach’s (2006) research into GBV in schools in Zimbabwe, Malawi, and Ghana found that it was common for educators to dismiss and trivialise sexualised forms of violence as inherent to male–female relations, and that this acceptance worked to normalise aggressive behaviour by boys as an inevitable part of the school system [17]. Muhanguzi (2011) found that hegemonic expressions of male power are prevalent within Ugandan secondary schools, and that this works to normalise the victimisation and subordination of young women [18].

### 1.1. Prevention Education

Early adolescence is not only a time of increased vulnerability, but it is also a key time to provide prevention education. A systematic review of 82 studies found that interpersonal influences on gender attitudes intensify during early adolescence (10–14), and that during this life stage, boys can encounter heightened pressures to demonstrate their strength, dominance, and sexual prowess, and girls encounter greater pressures to appear attractive, whilst suitably moderating any expression of their sexuality [19]. A self-report prevalence study with 1094 Ethiopian male students aged from early adolescence to 20 years found that those over 15 years perpetrated significantly higher rates of all forms of GBV as compared with those under 15 years [20]. During adolescence, the intersection of peer, family, and cultural norms can heighten and play out with boys and girls reinforcing or policing these gender norms within their friendship groups as well as within their cross-gender relationships. As such, it is recommended that school-based prevention education programs target interpersonal relationships as the key site of influence [19].

Evidence is available about the positive contributions that can be made via comprehensive school-based ‘respectful relationships’ programs that combine a focus on gender equality and the prevention of GBV. A meta-analysis of programs conducted in high income countries found that participating students had a better knowledge of what constituted GBV, attitudes less tolerant of gender-based violence, and reported lower rates of violence perpetration and victimisation [21]. A systematic review of comprehensive sexuality education (CSE) programs that contained content addressing gender and power drew on studies from 15 countries, including six from within Africa. A major finding was that dialogic approaches that fostered critical thinking about gender and power relations and contributed to the growth of positive teacher and peer relationships were more likely to lead to student empowerment in addressing gender inequities [22].

While there is a large body of research that argues the merits of student voices improving school policy and practice [23,24,25], less is known about how students perceive GBV prevention programs as research into such program provision has commonly focused on tracing the impacts on students’ knowledge, attitudes, and behaviours via the use of quantitative methods [22], rather than on inviting students’ perceptions of the value of the program and content. For students’ voices to be heard, they need to be afforded adequate opportunities to share their perspectives, and to be positioned as contributing [26,27,28]. A qualitative Scottish study that invited secondary school students to share their views as to the relevance and suitability of a GBV prevention program demonstrated that students could provide valuable insights into those aspects of the program they found most relevant, citing the contribution of dialogic learning activities and the focus on skills required for positive relationships [26]. From a more critical perspective, they also recommended the inclusion of more contextually and culturally relevant scenarios, and the use of additional scenarios that addressed the more minor or routinely experienced forms of gender-based harassment that they encountered in daily life. Similarly, in another qualitative study with students from the upper levels of a primary school in Botswana, students advised that they wanted to be able to talk more openly in their classrooms about issues related to sexuality and that they would prefer the use of a relationship-centric approach, rather than the more information-centric approach that they had experienced [29]. These two examples highlight the contribution that can be made to program evaluation and refinement via the inclusion of student voice within the evaluation methodology.

Given the relative dearth of student perceptions of school programs addressing the prevention of GBV and the recognised need for further research conducted across varied cultural contexts within low-income countries [19], it is here that this study seeks to contribute. It reports on the use of a student-centric approach to the analysis of data collected from 92 schools across the three countries in the East and Southern Africa region. Zambia, Eswatini, and Tanzania piloted UNESCO’s Connect with Respect: Preventing gender-based violence in schools (CWR) program, a prevention education program designed for use with students aged 12–15 years [1].

The intervention focuses on advancing interpersonal skills and the prevention of school-related GBV. It includes learning activities grouped into seven key topic areas addressing gender norms, human rights, positive cultural role models, understanding of the types and impacts of peer perpetrated GBV, communication skills for respectful relationships, peer support skills for bystanders witnessing forms of peer-perpetrated gender-based violence, and help-seeking knowledge and skills, as well as additional activities through which to contribute to a whole school approach. (See Table 1 for overview of the CWR topics and learning activities.)

The theory of change and instructional design of the program is strongly informed by research into effective approaches to social and emotional learning (SEL) and to the prevention of GBV. The six learning activities (or lessons) within Topic 1: Gender and equality use dialogic approaches to evoke critical thinking about gender norms, and the ways in which they can have limiting or harmful outcomes for both males and females. This is followed by five learning activities in Topic 2: Gender equality and positive role models. This topic combines a focus on human rights, and on the ways in which personal and cultural strengths and learnings from positive role models can be drawn on to resist negative social pressures relating to gender inequality. The approach in these two topics is informed by research that shows that effective respectful relationships programs use rights-based approaches, engage students in critical thinking about the social, cultural, and institutional influences on gender norms, and provide opportunities to build social skills via engagement with relevant scenarios and rehearsal strategies [30,31,32].

The following four learning activities in Topic 3: Raising awareness about gender-based violence focus on developing a shared understanding of the different forms that violence can take, including verbal, physical, sexual, and psychological, and on developing empathy for the negative effects on targets and observers. Students then progress to Topic 4: A focus on prevention of school-related gender-based violence. Here, the four scenario-based learning activities focus on strategies that peers can use to resist peer pressure to participate in violence, and to provide peer support and peer referral for those who are victimised. The learning activities in Topics 3 and 4 are informed by bullying prevention research that identifies the importance of approaches that address the operations of power relations within situations of interpersonal violence, and that use comprehensive definitions of violence that include verbal, sexual, financial, and psychological forms as well as physical violence [33,34,35]. Topics 5, 6, and 7 focus on developing the skills for respectful relationships, including friendship and peer support skills, skills for respectful and assertive communication, and help-seeking skills. They provide a combination of applied learning activities using small group problem-solving discussions around key scenarios, and role-play activities to practice ways to take problem-solving advice into action. The 12 learning activities in these topics are strongly informed by the large body of research investigating SEL, and the importance of using activities that include learning activities designed to advance peer support, problem-solving, and communication skills and are also informed by research into SEL [36]. Comprehensive SEL programs can lead to reductions in bullying, improved mental health, academic attainment, and school connectedness [37,38], as well as to reduced rates of cyberbullying, homophobic teasing, and sexual harassment [39]. The 31 learning activities across the program provide opportunities for collaborative learning and this is in line with research that demonstrates that collaborative learning activities are key to the effectiveness of wellbeing education programs [40] and that students prefer instructional approaches that provide them with opportunities to engage in dialogue about issues that are relevant to their lives and relationships [29]. (Further information about the theory of change informing the instructional design is available in the introductory section of the program itself [1,41]).

Program implementation is supported by a 5-day professional learning program for teachers that takes a participatory approach, providing opportunities to sample and discuss the learning activities designed for the students, along with a focus on the use of positive discipline strategies [42].

The CWR program was originally developed for students aged 12–15 years in the Asia-Pacific region, and ensuring a program is sensitive to the local context is important for intervention success. The misalignment of program content with the socio-cultural context can inhibit a program in achieving its objectives [27] and teachers may choose to omit topics that they believe to be incompatible with their comfort level or the values and traditions of their community. For example, Zulu et al. (2019) interviewed 18 teachers from rural schools in Zambia about their delivery of CSE and found that teachers commonly left out content they believed was not relevant to their students and/or conflicted with community norms [43]. Accordingly, a consultative process was used to modify the CWR program for use in African countries. A 5-day cultural consultation workshop led by the first author provided an opportunity for participants to sample a range of learning activities from the program and to contribute to revisions to ensure local relevance and cultural suitability [42].

### 1.2. Program Suitability and Adaptation

Efforts towards contextual and cultural adaptation are important, as is investment in professional learning for teachers. Teachers, particularly those in settings with cultural norms and traditions that conflict with some program content, find gender equality and GBV to be sensitive topics, reporting concerns about resistance and about triggering distress as they openly talk about silenced and normalised practices [29,44]. They may find that despite stating belief in the notion of gender equality and seeking greater freedom from restrictive gender norms, their colleagues and students may endorse beliefs in male superiority and express belief in the acceptability of certain forms of interpersonal violence [19], as may the broader society surrounding the school [27]. Teachers have also reported that teaching about GBV can entail significant investment of emotional, political, and pedagogical labour on their part [44]. Emotional labour is called for as the topic can bring up distress in their own lives as well as concern that the subject matter might trouble those students who have been victimised. Political labour is exerted in the face of backlash or resistance, as this type of education constitutes an effort towards broader social change. Pedagogical labour is required as teachers are called on to manage peer to peer interactions within collaborative and dialogic learning activities. Overall, managing a tension between the acceptability of dominant norms and the desire for change can present a significant challenge for teachers [45].

## 2. Materials and Methods

In order to focus on student perspectives, this paper reports on a sub-set of data generated within a broader mixed-method study. The wider study included the use of pre- and post-test quantitative data to examine the extent to which participation in the program led to improved peer relationships, reduced rates of harassment within and between gender groups, and stronger intentions to help-seek in relation to GBV. These results are reported elsewhere [41].

### 2.1. Procedure and Participants

The mixed-methods research project aimed to determine the extent to which the CWR program was implemented across the schools in Zambia, Eswatini, and Tanzania, using pre- and post-surveys to investigate the factors that constrained and enabled provision, and to track the impacts of the program on students’ experiences of peer-perpetrated bullying and GBV, their bystander capabilities, and help-seeking intentions. Via the qualitative components of the research, it also used interviews and focus groups to seek teacher and student perceptions about the contribution of the program.

With support from UNESCO and the various Ministries of Education in the participating countries, a 5-day teacher training was provided for all teachers. The CWR program was then delivered by teachers in 92 schools, including 24 schools in Zambia (17 primary and 7 secondary), 18 secondary schools in Eswatini, and 50 schools (20 primary and 30 secondary) in Tanzania.

The timing of program provision and data collection differed in the three contexts. The Eswatini delivery occurred in October 2019 and was truncated due to a late start, close to the end of the school year, leaving insufficient time for delivery. The Zambia delivery took place from August to November 2019. The Tanzania delivery was provided in March 2020, and from July to November 2020. Delivery was interrupted by three months due to school closures during the COVID-19 pandemic.

### 2.2. Instruments

In-country monitoring and evaluation teams were employed in each of the three countries to collect the data pre- and post-implementation using instruments that had earlier been developed by the lead author and colleagues. All data collection tools were given approval by the ethics committee at the University of Melbourne.

Student surveys were distributed pre- and post-program implementation. The surveys examined changes in student knowledge, attitudes, and experiences within peer relationships, and their knowledge and intentions in relation to help-seeking and peer support in situations involving GBV. Additional survey questions were used at post to investigate student experience of the intervention and to seek their recommendations about future provision. In total, there were 64 questions in the pre-implementation survey and 75 questions in the post-implementation survey, chiefly in multiple choice formats, using Likert scales.

Students from each school were invited to take part in single-sex focus group discussions. Two to four focus groups were conducted in each school, with separate groups for 4–8 girls and 4–8 boys. Focus groups explored student experiences relating to the influence of the program on their relationships with peers, their attitudes towards help-seeking, and their experience and feedback about the program.

Implementation surveys were completed by teachers across the duration of program delivery via a tool that collected data about whether each learning activity was provided in part, in full, or not at all, along with detail about key reasons for omission or modification. One on one teacher interviews were conducted in each school post-program delivery.

### 2.3. Data Set

Overall, 9089 students completed the pre-implementation surveys, and 9090 students responded at post. A total of 1069 students participated in the focus groups. Program monitoring surveys were received from 286 teachers (see Table 2 for detail).

### 2.4. Data Analysis

The monitoring and evaluation teams from each of the three countries shared their quantitative data with the authors in excel (Zambia) and SPSS (Eswatini, Tanzania) formats, and via summary of responses to the student focus groups. Data were collected in English in Zambia and Eswatini, and in Swahili in Tanzania, and provided in translation to the authors. On receipt of the survey data, it was cleaned to ensure all the variables had the same names, labels, values, and codes. The datasets were then merged into one SPSS file. Descriptive analyses were conducted using SPSS 26, with student responses calculated and compared between pre- and post-implementation.

Student focus group responses were summarised at point of collection by notetakers, and the country data were provided to the authors via thematic summary around the key questions asked in the focus groups. The summaries of qualitative data received by the research team were further analysed using thematic analysis to identify key patterns emerging from the data.

The monitoring and evaluation teams provided descriptive results of teacher monitoring data collected to provide an overview of program implementation and modification. The percentage of teachers delivering each of the learning activities within the seven topic areas was calculated by adding total percentages of delivery of each activity in the topic (delivered either in part or in full) divided by number of activities in the topic area. A summary of implementation data is presented in Table 3. It shows there were varying levels of program implementation across countries. Most teachers in Zambia (around 90%) and Tanzania (80–90%) delivered all learning activities in the seven topic areas either in full or in part. However, while most teachers in Eswatini delivered all learning activities within Topic One (74%), due to implementation challenges in relation to late timing in the school year, less than a third provided all activities for Topics Two (28%) and Topic Three (26%), and only 4% provided activities within Topic Four. There was no provision of Topics Five to Seven.

The key reasons given by teachers in Zambia and Tanzania for modification of the learning activities were lack of time, challenges in managing student behaviour, and the sensitivity of the material. The reason for partial provision of the program in Eswatini was due to late timing in the school year (commencing in October) leaving insufficient time for program provision, with only one month available before commencement of examinations period.

### 2.5. Data Limitations

There are a number of limitations in relation to the data that need to be taken into account when interpreting findings. First, data collection tools were translated into Swahili in Tanzania and then the results were back-translated, which could lead to inconsistency in translations of texts and content. In addition, there were instances in which the number of students completing the post-implementation survey exceeded the number of participants in the pre-implementation survey in Eswatini and Tanzania. Therefore, conclusions reached about shifts in student attitudes and behaviour might not fully represent the students participating in the pre-implementation survey.

Further, the teacher-monitoring survey asks if teachers completed each of the learning activities in full, in part, or not at all, and to select key reasons for modification or omission. These self-report data were reliant on teacher accuracy of recall, and it is possible that some records were not completed close enough to the event to ensure accuracy. Another data-related consideration pertains to the focus group discussion data. Focus groups were not recorded. Notetakers made summaries at the point of collection. The data provided by the evaluation teams to the authors came in the form of descriptions of key themes with some supporting quotes. Thus, one needs to be cognisant of the limits of the conclusions drawn from the focus group data, which had already been summarised both during and following the point of collection.

## 3. Results

In line with the focus of this paper, we discuss findings from the analysis of a sub-set of questions in the post-implementation student survey in which students were asked to evaluate the contribution of the program. We then interthread the focus group data to shed additional light on the ways in which students experienced the program as contributing towards improved peer relationships and positive shifts in gender-related behaviours, as well as their knowledge and intentions to seek help or support affected peers if encountering GBV. A report providing the analysis of the broader student survey data is published elsewhere [41]. In brief, it shows that the program achieved a reduction in reports of unwanted sexual comments and sexual touch by peers, reductions in the rates of negative bystander behaviours such as laughing, and an increase in positive bystander behaviours such as referral to a teacher among students if they witness GBV. There was also an increase in the proportion of students who reported that they knew how to seek help for those impacted by GBV as well as those who indicated that they would seek help on their own behalf if affected by GBV at school.

### 3.1. Student Recommendations about Future Provision of the Intervention

In order to gain a sense of the importance that students placed on efforts to advance gender equality and prevent GBV, questions were included in the endpoint survey to ask students whether they thought all schools should provide an intervention of this nature. A large majority of students in each of the countries selected ‘yes’, from yes/no/not sure as options (Eswatini: 84%; Zambia: 90%; Tanzania: 96%) (See Table 4). This validation likely indicates that students found the program reflective of their civic values and relevant to their social and cultural contexts.

In both Eswatini and Zambia, girls were more likely to recommend general provision than boys, whilst response rates were similar for boys and girls in Tanzania (Eswatini: M: 79%, F: 88%; Zambia: M: 87%, F: 92%; Tanzania: M 95%, F 96%). The stronger endorsement by girls fits a recognised patten whereby studies find that adolescent girls are more likely than boys to endorse programs addressing gender equality [18]. This may indicate that, due to higher rates of exposure to harassment and other forms of gender inequality, girls see a greater need for programming efforts.

Focus group data gathered from students shed further light on their views about the importance of wider provision as students in each of the focus groups recommended that other classes and schools should be providing the program. This endorsement was also echoed with requests to do further work of this nature in their own classes. Students in Zambia felt their class would benefit from doing more lessons because ‘we need to practice what we’ve learned’, and it will help to ‘reduce cases of GBV in the future in our schools and community’. Similarly, students in Eswatini thought that students in their class would benefit from more classes about GBV prevention and ‘there needs to be more than one period a week, so you don’t forget’. They suggested that more classes would increase harmony, reduce conflict, help those afraid to speak out, reduce name calling, increase bonding, and help them to spread information to their families and to other schools. Some pointed to the unique opportunity the classes provided to have conversations about issues with which they might not otherwise engage, commenting that they did not usually discuss such issues as growing up and sexual relations at home. Students in Tanzania made similar recommendations. All focus group respondents in this context felt their class would benefit from doing more lessons about how to have good relationships with others, so as ‘to know more about different techniques of preventing gender violence and the measures to take once they spot violence or when subjected to violence.’ One student noted, ‘we would like Connect with Respect to be included in the curriculum just like other subjects’.

Awareness of the need to also improve community attitudes was also an overarching thread in these recommendations about wider provision. Zambian students noted that that GBV ‘affects everyone from home and school’, ‘that GBV can be prevented if people are aware of it and its effects’, and that we want to ‘reduce cases of GBV in the future in our schools and community’. Similarly, a Tanzanian student noted, ‘it is not enough for us to learn, if other schools do not [teach this] since the problem will still be there’.

### 3.2. Student Views about Whether the Program Improved Their Relationship Skills

Students were also asked in the post-survey about the extent to which they believed doing the lessons had improved their relationship skills, responding on a 5-point Likert scale (see Table 5). Responses were very positive, and students who selected somewhat, mostly, or always as their response ranged from Eswatini 75%, Zambia 79%, and Tanzania 97%. Differences were not strongly marked along gender lines. There were equal responses from the genders in Tanzania, boys’ responses were higher than those of girls in Eswatini (M: 77%, F: 73%), and girls in Zambia were somewhat stronger in their endorsement than boys (M 78%, F 80%).

In the focus groups, the students reported that the process of working together on the collaborative tasks led to improved relationships with their classmates. As one Tanzanian student noted, ‘the education on respectful relationships has helped us live like sisters and brothers.’ This reference to increased harmony and connectedness was echoed widely. All of the focus group respondents in Zambia agreed that the lessons had helped to improve classroom relationships. They referred to increased respect for each other’s views, and less fighting, teasing, and abuse between classmates. One student noted ‘there has been a tremendous improvement in the way we relate in class’ because students are now ‘able to see each other as equal and share ideas as well as work together’. Others noted that ‘our class already has changed for the better’, ‘the lessons are helping managing misunderstanding and conflicts’, and ‘we can even help each other with schoolwork as good friends’.

The Eswatini students appreciated the use of games as making their interactions enjoyable as well as carrying clear messages about respect. They observed increased bonding between classmates and noted that some relationships had improved between boys and girls, with boys now being more considerate of girls. However, others noted that improving male–female relationships will take more time as ‘not all learners have changed their behaviour’, ‘some boys still don’t want to sweep the classroom’, and some boys still ‘find it difficult to befriend those who are gay’. Others noted that sharing feelings required trust and that this would take time to develop as ‘some people spread lies about those who shared feelings’, and some still fear being judged by their peers. It could be noted here that the Eswatini students were not provided with the last three topic areas of the program, which chiefly focused on communicating about emotions, along with peer support and help-seeking skills.

Tanzanian students pointed to the contribution of collaborative learning and, in particular, that working in mixed-sex groups ‘made us familiarise with one another even out of school’. They found that the ‘language we use has also changed’ in the way we speak to each other. They felt that they now behaved better in class, respected each other more, and had better relationships with both peers and teachers.

This is a relevant finding as numerous studies conducted with young people find that they want to know more about how to conduct healthy interpersonal relationships based on respect [46,47], and to be provided with opportunities to develop their skills in communicating effectively within peer relationships [48,49,50]. The views that students expressed about the importance of the collaborative learning strategies are also relevant as wider research shows that collaborative learning strategies are fundamental to the effectiveness of relationship education programs [40,51] as they assist students in rehearsing and developing the communicative skills that they can employ in their lives [21,52].

### 3.3. Student Rating of the Contribution of Program Components

Students were also asked in the end-point survey to rate how useful they found key aspects of the program. Their rankings showed the strongest validation for learning about gender equality and human rights (with 88% of students rating this program component as useful, very useful, or extremely useful). The next highest rating by 84% of students was for learning how to get help for those affected by violence, followed by an equal rating by 79% of students for understanding and communicating about their emotions, and strategies to avoid joining in with bullying and harassment. Only a slightly smaller proportion of students (78%) appraised as useful, very useful, or extremely useful the program components focused on understanding the effects of GBV (see Table 6).

Responses varied somewhat by gender, with a greater proportion of girls positively appraising usefulness than boys. Ratings varied more substantially by country of implementation (see Table 7), with the strongest response in Tanzania, followed by Zambia and Eswatini. However, the relative rankings of usefulness were closely consistent by country.

Focus groups not only provided an opportunity to deepen insight into what students found to be useful, but also demonstrated the interconnected nature of different program components, in that the focus on understanding and communicating about feelings played a role in heightening awareness of the impacts of GBV, and taken together, both played a role in the capacity to provide peer support, intervene as a bystander, and to desist from joining in with bullying or harassment. Taken together, the students valued the program as providing a relationship-centric approach to the prevention of GBV, oriented by a rights-based awareness of the importance of gender equality and respect for all.

### 3.4. Students Endorse the Focus on Human Rights and Gender Equality

The strong endorsement of the focus on learning about gender equality and human rights was not only evident in the survey data, but also in the focus group data. (Rating for the usefulness of this aspect of the program ranged from Zambia: 82% of students, Eswatini: 87% of students, and Tanzania: 95% of students). Both boys and girls in Tanzania maintained that learning about fairness and human rights had heightened their awareness of inequity and opened new possibilities for them. One student maintained that ‘I now understand even myself as a boy, when I get home, I have to do domestic chores without discrimination’. In Eswatini, both boys and girls said they thought it was important to learn about fairness and rights so as to be more aware of how to treat each other equally and not to infringe on people’s rights. Girls in Eswatini noted that they felt empowered by the acknowledgments that boys and girls are capable of doing all jobs and roles and found that this affirmation improved their self-esteem and self-respect. Learning about gender stereotypes and gender equality also fostered their hope that this would help to ‘break the cycle of associating females only with domestic chores’. However, as previously noted, some girls observed there remained some boys who did not wish to do ‘domestic’ chores and were challenged in accepting those who were gay. Some of the boys acknowledged that this kind of fairness was difficult for them to put into practice but that the lessons had helped to raise their awareness of inequality.

This increased awareness of inequities based on gendered expectations is consistent with a wider body of research that demonstrates those with rights-affirming attitudes are less likely to engage in gender-based violence [53,54].

### 3.5. Learning to ‘Identify a Violation’ of Rights

Closely allied to the focus on human rights, students also found it very useful to focus on the different ways in which GBV could manifest. Zambian students found that it was useful to focus on the different the types of violence, such as verbal and sexual, as well as physical, and to as focusing on avoiding negative types of by-stander behaviour such as laughter or watching on. They pointed to the importance of this naming power, noting, ‘we have learnt a lot of things like breaking the silence by reporting gender-based violent acts, some of which we suffered without knowing whether they were gender-based violent acts, for example someone may touch you without your consent.’ One Tanzanian student pointed out that they can now ‘identify different forms of violence in school’, and another noted that this has helped them to ‘identify a violation’ of their rights. Another noted that it made them aware that ‘gender violence makes an individual unconfident, unhappy and consequently fail to realise one’s dreams and also lose one’s life in some cases.’ The association between an awareness of understanding what constitutes violence and understanding when an infringement of rights had occurred is evident here. Broader prevalence data revealed via the Violence Against Children studies in these countries shows that one of the key reasons for not reporting sexual abuse is not knowing that it was a reportable problem [55,56,57].

### 3.6. Learning to Express Feelings ‘Without Any Fear’

Students observed strong connections between the capacity to recognise and express one’s own emotions, and the capacity to empathise with those affected by GBV. The post-survey results showed that around three-quarters of students found this aspect of the program useful (Zambia: 74% of students, Eswatini: 76% of students, and Tanzania: 88% of students). Many Tanzanian students said that the focus on skills for communicating about their feelings and needs had helped them to build confidence, self-awareness, and their capacity ‘to express their feelings without any fear’ particularly when intervening as a bystander. Zambian students similarly noted the importance of learning how to communicate their feelings, valuing this a key ‘life skill’. Some Eswatini students saw a link between learning how to express emotions and the prevention of depression. Boys commented that communicating about feelings helped to ‘heal the soul’, promote trust and empathy, and helped them to better handle their feelings. Girls noted that ‘you reduce stress levels by speaking out’, and you get ‘relief from what was eating [you] inside’, from pain and anger. Some believed that this work would reduce suicidal ideation, lead to better decision making, encourage friendship, and create better bonds between students.

This is an encouraging finding as research into the provision SEL programs demonstrates that the development of such self-awareness, social awareness, and the capacity to understand and express emotions within respectful relationships is associated with improvements in peer relationships, reductions in bullying and harassment amongst peers, and a reduction in mental health problems such as anxiety and depression [37,58,59,60].

### 3.7. Developing ‘Courage to Reject GBV and Raise Your Voice to Eradicate Violence’

Students also endorsed the program’s focus on how to avoid joining in with the negative treatment of peers and the opportunity to discuss the challenges associated with becoming a proactive bystander, as well as to rehearse possible strategies (Zambia: 72%, Eswatini: 79%, and Tanzania: 87%). In Zambia, both boys and girls found that the program helped them ‘to reduce fights and misunderstandings’, to ‘learn self-control’ to ‘avoid teasing and to encourage each other’. Tanzanian students explained ‘we understood challenges that a victim of gender violence goes through’, and this involved learning ‘the signs to look for, and to feel empathy’, which was a precursor to supportive action. In Eswatini, both boys and girls pointed to the importance of activating a sense of compassion, as well as teaching how to support victims, noting that this learning helps both victims and bystanders. Students from Eswatini also noted the courage needed to intervene as ‘learners need to be bold in order to help others’. They talked about how the program built their courage to speak out and provide help, noting ‘without courage it is difficult to help. You may see a victim but fear to help as people may connect you with him/her, but with this education, confidence does it all’. Another noted that it takes ‘courage to reject GBV and raise your voice to eradicate violence’.

Overall, students found it useful to explore and rehearse reactions and possible bystander interventions in relation to different manifestations of violence. Given the chiefly social nature of peer-perpetrated bullying and GBV amongst adolescents, it is logical that students appreciate a focus on strategies, skills, and strengths needed for proactive bystander responses to these forms of violence. Research investigating bullying prevention finds that approaches that focus on promoting positive responses from bystanders are more effective than moralistic approaches, particularly when working with adolescents as they are more affected by the complexities of managing peer connectedness and membership of friendship groups than younger children [26]. Additionally, wider studies have shown that as students enter puberty, their identities and cross-gender relationships become more sexualised in the eyes of peers, parents, and community members, and young people experience greater pressure to perform into gender norms [19]. Such pressure can lead to both the perpetration and/or acceptance of experienced and witnessed violence.

Research also suggests that it is important to equip bystanders to challenge the sexist, homophobic, and transphobic attitudes and actions that can lead to forms of GBV [55,56,57]. Effective learning activities focus on developing positive strategies for intervention, and on advancing the skills needed to carry out these strategies within everyday social contexts [61].

### 3.8. It Was ‘Useful for Awareness and Knowing How and Who to Report GBV to’

The student survey responses showed that the majority found it useful to learn about how to get help for those affected by GBV (rated as useful, very useful, or extremely useful by 80% of Zambian students, 82% of those from Eswatini, and 91% of those from Tanzania). The learning activities focused on how to provide peer support and peer referral as well as a focus on help-seeking on one’s own behalf. The activities included the use of role-play as a rehearsal strategy and information about help-sources in schools and communities. As one Tanzanian student noted, there were interconnections between naming acts of violence, recognising them as an abuse of rights, desisting from perpetration, and stepping up as a bystander. ‘(At first) they didn’t know that those deeds were wrong so, they too joined in, but after receiving education from this program, they changed and are now able to help the victim’. Others in the focus group affirmed that the activities ‘have been useful for awareness and knowing how and who to report GBV to, e.g., the gender desk at the police station’. Similarly, Zambian students noted the connection between empathy and the capacity to provide peer support. All students felt it was important to learn about how to be a good friend and help-seek for those classmates who have been affected by GBV as GBV ‘can happen to anyone’, and those affected by GBV ‘need support, feel lonely and isolated and might need our encouragement in order to continue with their education’. For this it was also ‘important to know what kind of help is available and where help can be found’.

Research demonstrates that help-seeking attitudes are influenced by gender norms, as well as cultural attitudes [62]. It is important, therefore, to normalise help-seeking and build the skills and knowledge that young people need to seek support. Further, encouraging and fostering help-seeking behaviours has been shown to be one way to improve mental health and wellbeing [63]. There are a number of psychosocial, cultural, and practical barriers that young people can face when it comes to seeking help in relation to sensitive issues pertaining to mental, social, sexual, and family distress [64]. Research demonstrates that it can be the people who most need help who are least likely to feel that they can enter a help-seeking pathway [65]. If they do seek help, they may be more likely to turn to peers, rather than to formal sources. Therefore, it is important to build skills for peer support and peer referral, as well as to normalise help-seeking as a legitimate action.

Implicit in the value students ascribed to learning about how to support peers is an underpinning ethic of care whereby community members feel a strong sense that action should be taken on behalf of those who have been subjected to violence. The translation of this ethic of care into compassionate action has been termed the work of collective ‘response-ability’ by scholars engaged in similar work designed to challenge forms of racism [66]. Given the shared, inherited, and ongoing co-constructed nature of gender norms, emphasis on the collective nature of ‘response-ability’ fits well with what the students have shown that they find useful—an interconnection between an applied rights-informed gender-equitable approach to understanding and expressing emotions within positive relationships, building their capacity to withstand invocations to enact negative treatment and advancing their confidence and capacity to befriend others, and provide help or support where needed.

## 4. Conclusions

Studies investigating the contribution of programs addressing the prevention of school-related GBV have rarely invited students themselves to evaluate the contribution of the program. This is despite research that shows young people can provide nuanced accounts of their needs, priorities, and relational concerns and are well-placed to comment on the effectiveness of programs used to address their needs [26,29,67]. There is, thus, a contribution that can be made by research that investigates what students find useful within programs provided for them.

This research found that students valued the opportunity to engage in education that included an emphasis on gender equality, the prevention of GBV, and skills for positive relationships. They found that this led to them becoming more aware of the importance of gender equality, the impact of violence on others, and of the strategies they could use to provide peer support and to engage in help-seeking. Further, due to the focus on communication, self-awareness, and peer support skills, they found that the program led to improvements in their own relationships, with this pointing to the importance of including a focus on SEL within programs addressing the prevention of school-related GBV. That the majority of students in this study recommended that similar programs be provided in all schools indicates a strong appreciation of relevance and utility.

Potentially, student voice data of this nature could be used to strengthen the case for the inclusion of this kind of programming within national curricula and school timetabling, and to allay teacher concerns about resistance. This is relevant as teacher anxieties regarding the possibility of alienating men and boys are commonly raised when schools are asked to embrace such programs [44,68] and can lead to teachers reverting to the provision of generic life skills and bullying-prevention programs and omitting learning activities that more specifically address the gendered nature of many forms of interpersonal violence [68]. Additionally, when education systems and schools see evidence of the positive impact in their own or similar contexts, they may be more likely to include such approaches in the official curriculum and ensure that training and resources are provided to support teachers in their implementation efforts [2].

Program outcomes and student validations of usefulness were stronger in Zambia and Tanzania than in Eswatini, where program provision was limited to around the first 20% of the program. The stronger endorsement by students in Tanzania may possibly reflect the most comprehensive delivery. However, it may also reflect the timing of the provision in that country, provided when students returned to school in July 2020 after a three-month absence due to school closures precipitated by the COVID-19 pandemic. Research investigating the impact of school-based social and emotional learning programs post-emergency shows that they make a significant contribution to student wellbeing and recovery post-disaster [69,70,71]. Following a prolonged absence, students can find it challenging to socially re-connect with peers and, for some students, anxiety increases as they return to school [72]. Programs that focus on teaching skills for positive relationships can thus stand to make a particular contribution at times of heightened social need.

There are implications for systems and schools regarding finding time in the curriculum and timetable for comprehensive provision. The problem of finding adequate time is well-identified in research investigating the implementation of wellbeing education in secondary schools [73], as is the need for consistent provision over an extended period [74]. A review of GBV prevention programs found that success factors in relation to programs delivered over two years to children in Pakistan and Afghanistan included that sufficient time was allowed for the development of critical reflection, problem-solving, communication, and collaboration skills supporting respectful non-violent behaviors [74].

A system and school-level commitment to providing teachers with guiding resources is important, as programs produce better outcomes when teachers are provided with access to training and to research-informed lesson plans [75,76,77,78] as well as guidance in relation to facilitating the collaborative learning activities that are central to such programs. This pedagogical support is important as omission of the collaborative learning activities is one of the most common areas of breakdown in fidelity at the point of provision of wellbeing education programs with a consequent reduction in program impact [79,80,81,82].

### Limitations and Need for Further Research

There are a number of limitations that preclude the generalisability of the findings of this study. The intervention was not provided in full in all settings, and whilst this may have led to the variability in the ratings of usefulness on the part of the students, it also may be an indicator of cultural or educational differences in the various schools or countries. Whilst the focus group data were consistent with the findings from the post-program student evaluation survey, and provided a degree of insight into what it was about the program that students found helpful within their peer relationships, the focus group data were provided in summary form, with a limited number of direct quotes. This limited the depth of analysis that could be conducted. Further quantitative research in full control trial conditions could provide a more rigorous examination of the influence of the program on attitudes, skills, and behaviour, and the intersections between implementation fidelity, gender, and perceived value. Further qualitative research could be deployed to explore student recommendations about program improvement and attunement to challenges associated with gender, social class, abilities, context, or culture. Further cultural consultation and research investigating contextual and methodological relevance is also warranted to inform the appraisal of suitability of the program for use in different contexts.

## Figures and Tables

**Table 1 ijerph-20-06498-t001:** Connect with Respect: Preventing gender-based violence in school program topics and activities.

Connect with Respect: Preventing Gender-Based Violence in School		
Part 1: Understanding Gender and Gender Equality	Part 2: Raising Awareness about GBV	Part 3: Developing Skills for Respectful Relationships
Topic 1: Gender and equality Activity 1: What is gender? Activity 2: Unpacking gender norms Activity 3: Messages about males and females in the media and literature Activity 4: Challenging negative gender norms Activity 5: Challenging myths Activity 6: Local leaders Topic 2: Gender equality and positive role-models Activity 1: Positive role models Activity 2: Fairness, equality, and human rights Activity 3: Human rights and gender equality in everyday moments Activity 4: Positive and negative uses of power Activity 5: Differences and discrimination	Topic 3: Awareness of GBV Activity 1: What is violence? Activity 2: What is GBV? Activity 3: Effects of GBV on males and females Activity 4: Negative health impacts of gender norms Topic 4: A focus on school-related gender-based violence Activity 1: School mapping of GBV Activity 2: Positive rules for the safe learning space Activity 3: Empathy, imagination and hidden emotions Activity 4: Making an apology	Topic 5: Communication skills for respectful relationships Activity 1: What good friends do Activity 2: Respectful relationships between males and females Activity 3: Introducing assertiveness Activity 4: Using emotions statements in respectful relationships Topic 6: Skills for people who witness violence Activity 1: Effects on the witness Activity 2: Building support strategies Activity 3: I want to do something to help! Activity 4: Active listening for peer support Topic 7: Help-seeking and peer support skills Activity 1: When and if to seek help Activity 2: Where to go for help Activity 3: Overcoming resistance to help-seeking Activity 4: Messages of support

**Table 2 ijerph-20-06498-t002:** Overview of sample size.

	Zambia	Eswatini	Tanzania
Student pre-implementation survey (N = 9089)	2400 students 1053 (44%) boys 1344 (56%) girls 2 indicated themselves as other 1 did not specify their gender	1661 students 797 (48%) boys 801 (48%) girls 63 (4%) did not specify their gender	4655 students 2131 (46%) boys 2513 (54%) girls 11 did not specify their gender
Student post-implementation survey (N = 9090)	2196 students 945 (43%) boys 1235 (56%) girls 2 selected ‘other’ 14 did not specify their gender	1764 students 788 (45%) boys 889 (50%) girls 87 (5%) did not specify their gender	4798 students 1977 (41%) boys 2797 (58%) girls 24 did not specify their gender
Student focus group respondents N = 1069 9M = 509, F = 560)	384 (M = 192, F = 192)	217 (M = 109, F = 108)	468 (M = 208, F = 260)
Program monitoring surveys N = 286 teachers	Zambia N = 84	Eswatini N = 52	Tanzania N = 150

**Table 3 ijerph-20-06498-t003:** Proportion of teachers in each country providing all learning activities in the seven topic areas either in part or in full.

Topic Area	Zambia	Eswatini	Tanzania
Topic 1: Gender and equality	97%	74%	84%
Topic 2: Gender equality and positive role models	90%	28%	83%
Topic 3: Awareness of gender-based violence	95%	26%	91%
Topic 4: A focus on school-related gender-based violence	95%	4%	88%
Topic 5: Communication skills for respectful relationships	97%	0	87%
Topic 6: Skills for people who witness violence	93%	0	89%
Topic 7: Help-seeking and peer support skills	93%	0	84%

**Table 4 ijerph-20-06498-t004:** Proportion of students who answered at end point that all schools should teach about the prevention of gender-based violence.

Site	Yes			No			Not Sure		
	Boys	Girls	Total	Boys	Girls	Total	Boys	Girls	Total
Zambia	87%	92%	90%	8%	4%	6%	5%	4%	5%
Eswatini	79%	88%	84%	8%	4%	6%	13%	9%	11%
Tanzania	95%	96%	96%	2%	2%	2%	3%	2%	3%

**Table 5 ijerph-20-06498-t005:** Student responses towards the Connect with Respect program: Doing the Connect with Respect lessons improved my relationship skills.

Site	Not at All/A Little			Somewhat/Mostly/Always		
	Boys	Girls	Total	Boys	Girls	Total
Zambia	22%	20%	21%	78%	80%	79%
Eswatini	23%	28%	26%	77%	73%	75%
Tanzania	3%	3%	3%	97%	97%	97%

**Table 6 ijerph-20-06498-t006:** Percentage of students selecting program component as useful, very useful, or extremely useful.

Components	M	F	Total
1. Learning about human rights for males and females	86%	89%	88%
2. How to get help for those affected	83%	86%	84%
3. Understanding and communicating about feelings	78%	81%	79%
4. How to avoid joining in with bullying	77%	80%	79%
5. Understanding effects of gender-based violence	78%	82%	78%

**Table 7 ijerph-20-06498-t007:** Proportion of students rating elements of the Connect with Respect program as useful, very useful, or extremely useful.

Percentage of Students Selecting Useful, Very Useful or Extremely Useful	Zambia			Eswatini			Tanzania		
	M	F	Total	M	F	Total	M	F	Total
Learning about human rights for males and females	79%	84%	82%	85%	89%	87%	95%	95%	95%
How to get help for those affected	78%	81%	80%	79%	85%	82%	91%	91%	91%
Understanding and communicating about feelings	72%	76%	74%	76%	77%	76%	88%	89%	88%
How to avoid joining in with bullying	70%	73%	72%	77%	81%	79%	86%	87%	87%
Understanding effects of gender-based violence	77%	82%	79%	85%	89%	82%	73%	75%	74%

## Data Availability

Data on the wider results of the study are available within an open access research report.

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
