# Peer review of "A Student-Centric Evaluation of a Program Addressing Prevention of Gender-Based Violence in Three African Countries"

_ijerph, 2023, doi:10.3390/ijerph20156498_

Round 1

Reviewer 1 Report

Thank you so much for the work well done. What an insightful information supported by relevant sources?

Author Response

No revisions to respond to.

Reviewer 2 Report

Dear Authors, 

First of all, I would like to thank you for reading and evaluation opportunity on such an important topic.

The literature review of the different elements of the research is sufficient, as is the description of the data collection and analysis. What needs to be expanded is the explanation of the internal content of the modules developed (whether they were based on scientific evidence or not). There also needs to be adequate modulation of the results based on the data: there was little recognized change in aspects of the program (less than 25% would not be high), this needs to be pointed out. This needs to be picked up in the conclusions, and the little result in this regard is strongly emphasized in the text. It should be exposed in the article, as a highlight, that the research does not reach results that would allow recommending transfer of the program to other countries. The research should be redone with greater control of the data and analysis. This should be made explicit in the article, but does not diminish the importance of the research, as it reveals elements to be taken into account in future research.  

Author Response

Thank you for the considered and thoughtful feedback. We have carefully reviewed this and have revised the manuscript accordingly.  

  1. Reviewer comment:

 What needs to be expanded is the explanation of the internal content of the modules developed (whether they were based on scientific evidence or not)

Response:

The evidence-base upon which the program was developed has been detailed. This revision in highlighted in Section 2: Prevention education.

With regard to expanding on the internal content of the modules, Table 1 in this section lists all the topics in the program. We believe this is sufficient for the reader to have an understanding of what students were taught. Further, if the article is published, the program will be accessible to those readers seeking more detail as it is open access.

  1. Reviewer comment:

There also needs to be adequate modulation of the results based on the data: there was little recognized change in aspects of the program (less than 25% would not be high), this needs to be pointed out. This needs to be picked up in the conclusions, and the little result in this regard is strongly emphasized in the text. It should be exposed in the article, as a highlight, that the research does not reach results that would allow recommending transfer of the program to other countries.

Response:

In response to these comments, we have revised removing the effort to focus both on the intervention impact and on analysis of the students’ perceptions in the one paper. Suggestions of transferability based on the results have been removed from the paper.

  1. Reviewer comment:

Research should be redone with greater control of the data and analysis. 

Response:

As noted, based on this thoughtful feedback, we have revised the paper to one that focusses on students’ perceptions of the value of the various components of the program. This is drawn from student responses to a sub-set of evaluation questions in the post-program survey and from student focus group data. Sections with analysis of program impact, using pre- and post-survey data, have been removed.

Reviewer 3 Report

Thank you for the opportunity to read this manuscript. I think the study is interesting and the authors' intention and goals are well-defined. The authors sought to examine the perceptions of students in three low-income countries in Africa regarding the importance of gender-based violence and the effects of interventions. I do not see major issues except the methodology and data analysis approaches the authors used:

(1) The study was able to collect massive (potentially highly valuable) quantitative data but why did the authors decide to conduct a simple statistical analysis of percentages and didn't go further to conduct a more sophisticated analysis of variables? Presenting only the percentages does not help with the scientific soundness of the results since they initially adopted a 'pre-post design'. I didn't seem to get the answer to the lingering question of "SO WHAT?" and "Why such a design?"... Especially see the next point

(2) I also found troubling the fact that "the trials were without the benefit of a control-trial design, and thus it is not possible to attribute changes entirely to the provision of the program". I see it as a (significant!) limitation.

The second limitation is also significant and somewhat negatively affects the strength of conclusions: "the study did not permit tracking of individuals from pre to post, as student identifiers were not included in the survey instruments". Why would you design it in such a way?

(3) A minor point: It would be appropriate to see a more elaborate discussion of what difference this study makes when comparing 'low-income' and 'medium-income' countries in Africa. In other words, in the implications, the authors should discuss the importance of examining 'income' as a significant variable in gender-based violence in a school context.  

Author Response

Thank you for the considered and thoughtful feedback. We have carefully reviewed this and have revised the manuscript accordingly.  

  1. Reviewer comment:
    1. The study was able to collect massive (potentially highly valuable) quantitative data but why did the authors decide to conduct a simple statistical analysis of percentages and didn't go further to conduct a more sophisticated analysis of variables? Presenting only the percentages does not help with the scientific soundness of the results since they initially adopted a 'pre-post design'. I didn't seem to get the answer to the lingering question of "SO WHAT?" and "Why such a design?"... Especially see the next point
    2. I also found troubling the fact that "the trials were without the benefit of a control-trial design, and thus it is not possible to attribute changes entirely to the provision of the program". I see it as a (significant!) limitation.
    3. The second limitation is also significant and somewhat negatively affects the strength of conclusions: "the study did not permit tracking of individuals from pre to post, as student identifiers were not included in the survey instruments". Why would you design it in such a way?

Response:

As these comments all speak to the quantitative component of the research design they will be addressed as a whole.

Based on this thoughtful feedback, we have revised the paper to one that focusses on students’ perceptions of the value of the various components of the program and their views as to the value of this type of program in their own and other school settings. This draws on a subset of student responses in the post-program survey about the sections of the education program they valued and from student focus group data which provides further detail of what was found by students to be useful. We have provided a thematic analysis of the student focus group data, and discussed this in relation to the ratings they gave to the program in the evaluation component of the end-point survey. This we believe caters to a recognised need for research which includes a focus on what student’s themselves find to be of value within prevention education.

Sections with analysis of program impact, using pre and post-survey data, have been removed and will be addressed in subsequent papers.

  1. Reviewer comment:

A minor point: It would be appropriate to see a more elaborate discussion of what difference this study makes when comparing 'low-income' and 'medium-income' countries in Africa. In other words, in the implications, the authors should discuss the importance of examining 'income' as a significant variable in gender-based violence in a school context.  

Response:

Based on this comment, we removed reference to income as this was not a focus of the revised paper.

Round 2

Reviewer 2 Report

When analyzing the remade text, it is observed:

1) In the Introduction, although there was no need, there was the addition of references on gender violence in the world and on the African continent. The references presented in the first version of the text already supported a consistent argumentation. The additions are not necessary.

2) In a different way, in the subsection entitled Prevention education, the insertions of related literature added important arguments about the theme and the relevance of the study.

3) The description of the program was not much expanded, and lacks articulation. In view of the additions made, there was a lack of comparison of each step of the Program with what other successful programs have shown to be successful in training for the prevention of violence in schools. The indication that it is necessary to show whether the internal content of the modules developed were based on scientific evidence or not was not met.

4) About the necessary modulation of results based on the data, what was pointed out in the review of the first version of the article (e.g., that there was little recognized change in aspects of the program (less than 25% would not be high), is still problematic in the second version. In the second version, weight was given to statements about what was revealed in the focus groups, with examples from either subject. It needs to be shown methodologically how the discourse data was analyzed. How many students agree with each category arising from the qualitative analysis.

5) The new drafting has softened the problems of approval and generalization that the article seeks to give to the program analyzed. But the data are still not consistent enough to allow such a claim.

6) For these reasons, I manifest again that it should be exposed in the article, as a highlight, that the research does not reach results that would allow us to recommend the transfer of the program to other countries. The same limits to the development of the research that appeared in the first version of the text remain, but the form of writing used in the second version minimizes them - which is quite worrying. I state again that, in the text, it would be important to make it explicit that the research should be redone with greater control of the data and the analysis. This should be made explicit in the article.

7) Programs that focus on violence prevention are very important and this research is part of the necessary production of knowledge on the subject. However, without revealing the limits of the attempt to transfer the project to African countries, as we indicated in the first opinion, it would be making a mistake, creating more problems than solutions in schools.

8) The elements brought by the voice of student participants what needs to be present in training programs for gender-based violence prevention in schools is quite interesting. I suggest that the authors address this in another article.

Author Response

We wish to thank Reviewer 2 and the Academic Editor for their further advice. We have made minor changes informed by this advice. (The changes are highlighted in track changes and marked in blue highlight.)

In the limitations section, we have added to the limitations listed, making clear that further research is warranted before presuming applicability to other settings or contexts, and that further research with full control trial conditions is warranted to more rigorously examine program outcomes in various settings in relation to fidelity, gender and other demographic differences. 

In response to the view that the description of the program was not much expanded, we have added more detail about how particular sections of the program are informed by the evidence-base, though we remain constrained by the length of the article about how much detail can be added here. We have also added a note that further detail of the theory of change is provided in the program itself, which is available open access.

In line with Reviewer 2’s advice, we note that the substantial changes to this paper now mean that it primarily deals with student perceptions about what they find useful education programs for gender-based violence prevention.

Reviewer 3 Report

Thank you for sharing this. I can see that the authors made significant changes. I think the manuscript can now be recommended for publication.

Author Response

(The authors gave the same response as above.)
